# New Insights into the Enhancement of Adventitious Root Formation Using N,N′-Bis(2,3-methylenedioxyphenyl)urea

**DOI:** 10.3390/plants12203610

**Published:** 2023-10-18

**Authors:** Ada Ricci, Eugenia Polverini, Stefano Bruno, Lucia Dramis, Daniela Ceresini, Antonio Scarano, Carmen Diaz-Sala

**Affiliations:** 1Dipartimento di Scienze Chimiche, della Vita e della Sostenibilità Ambientale, Università di Parma, Parco Area delle Scienze 11/A, 43124 Parma, Italy; 2Dipartimento di Scienze Matematiche, Fisiche e Informatiche, Università di Parma, Parco Area delle Scienze 7/A, 43124 Parma, Italy; 3Dipartimento di Scienze degli Alimenti e del Farmaco, Università di Parma, Parco Area delle Scienze 27/A, 43124 Parma, Italy; 4Departamento de Ciencias de la Vida, Universidad de Alcalá, 28871 Alcalá de Henares, Spain

**Keywords:** ABP1 auxin receptor, adventitious rooting, CKX inhibitor, urea derivative

## Abstract

Adventitious rooting is a process of postembryonic organogenesis strongly affected by endogenous and exogenous factors. Although adventitious rooting has been exploited in vegetative propagation programs for many plant species, it is a bottleneck for vegetative multiplication of difficult-to-root species, such as many woody species. The purpose of this research was to understand how N,N′-bis-(2,3-methylenedioxyphenyl)urea could exert its already reported adventitious rooting adjuvant activity, starting from the widely accepted knowledge that adventitious rooting is a hormonally tuned progressive process. Here, by using specific in vitro bioassays, histological analyses, molecular docking simulations and in vitro enzymatic bioassays, we have demonstrated that this urea derivative does not interfere with polar auxin transport; it inhibits cytokinin oxidase/dehydrogenase (CKX); and, possibly, it interacts with the apoplastic portion of the auxin receptor ABP1. As a consequence of this dual binding capacity, the lifespan of endogenous cytokinins could be locally increased and, at the same time, auxin signaling could be favored. This combination of effects could lead to a cell fate transition, which, in turn, could result in increased adventitious rooting.

## 1. Introduction

Adventitious roots are induced postembryonically from tissues other than those from which roots usually originate, due to the plasticity of plant cell development. This physiological process is either affected by endogenous factors related to the plant (i.e., genotype, maturation of the mother plant, nutritional conditions, hormonal balance), or is stimulated by stressful environmental conditions, such as wounding, flooding, etiolation, and nutrient deficiency, among others [1,2,3,4,5,6,7,8,9]. This natural ability allows individual plants with their own root system to be obtained from explants of stems and/or branches, so it has been commercially exploited for in vivo and/or in vitro vegetative propagation of ornamental, horticultural and forest tree species [10,11]. However, not all plant species readily form adventitious roots. Two main categories have been proposed, those that are easy and those that are difficult to root [12,13]. Thus, adventitious root formation is currently known as a bottleneck in the large-scale clonal propagation of many plant species, especially woody plants [14]. Many studies have been conducted on the basic nature of the process, and several horticultural practices have been developed to overcome this crucial step. From a physiological point of view, adventitious root formation is a progressive process that has been divided into induction (or de-differentiation), initiation (or re-differentiation) and, finally, expression phases, each with specific hormonal requirements [15,16,17]. Auxin is recognized as a key factor that stimulates and regulates adventitious root formation and development in competent cuttings of Gymnosperms and Angiosperms [17,18]. Other hormones, such as ethylene, gibberellic acid, jasmonates, abscisic acid, or cytokinins, are variously involved in the three successive and interdependent phases of adventitious rooting, some of them showing stimulating, and others inhibitory, effects [4,19]. It has been often thought that cytokinins counteract the adventitious-rooting process, but a more nuanced negative implication should be considered, as their effect differs according to the rooting stage. In fact, it has been reported that these cell division-promoting factors may favorthe formation of meristemoid niches, which are sensitive to the auxin stimulus, thus positively affecting the adventitious rooting process in an auxin/cytokinin crosstalk perspective [14,15,20].

Wounding or excision seems to be involved in the formation of local auxin maxima at the level of the wounded zone through basipetal polar auxin transport (PAT), thus creating the conditions for competent plant cells to respecificate their fate, de-differentiate, and, subsequently, re-differentiate to form a root meristem through coordinated cell divisions. As a result, a very common horticultural practice to stimulate adventitious root formation in the propagation of stem cuttings from difficult-to-root species is the basal application of exogenous auxin, usually indole-3-butyric acid (IBA) [6,21,22,23,24,25]. However, exogenous auxin supplementation alone can cause undesirable side-effects, such as the formation of extensive callus and/or the extrusion of stunted or malformed roots that hardly allow further successful in vivo acclimatization [14,26]. To reduce these deleterious effects and, at the same time, to improve adventitious rooting, different chemicals have been combined with auxin in exogenous supplementation [27,28,29]. In this regard, and as a continuation of previous publications [30,31,32,33,34,35,36], we report here new data obtained either via different in vitro bioassays or via computational studies of docking simulations on the adjuvant activity of N,N′-bis-(2,3-methylenedioxyphenyl)urea (2,3-MDPU, Figure 1) in the formation of adventitious roots.

The investigations carried out are the first attempt to verify if and how 2,3-MDPU interacts with the complex crosstalk between auxins and cytokinins that naturally regulates the adventitious-rooting process. To test whether its activity might be due to some type of interference with auxin efflux, *Malus pumila* Mill. slices and cuttings were treated with different combinations of auxin and/or 2,3-MDPU, either simultaneously or after supplementation with *N*-1-naphthylphthalamic acid (NPA). NPA is a well-known inhibitor of polar auxin transport, which blocks auxin efflux movement between cells, perturbs plant development and prevents the formation of the auxin maxima necessary for adventitious rooting [37,38,39,40,41]. In addition, apple cuttings supplemented initially with NPA, and, subsequently, with exogenous auxin and/or 2,3-MDPU were histologically analyzed to test whether the de novo organization of root meristems occurs as extremely localized areas of cell division due to the presence of 2,3-MDPU in the rooting medium, as previously reported in a distantly related woody Gymnosperm [32].

To further investigate and highlight the nature of the adjuvant activity of 2,3-MDPU on adventitious rooting, two approaches were followed.

First, an interaction with auxin receptor systems has been hypothesized. As 2,3-MDPU has not yet been shown to enter cells, our research has focused on auxin binding protein 1 (ABP1), whose apoplastic fraction favorably binds auxin at the acidic pH of the cell wall (ABPI localizes even in the endoplasmic reticulum, where the pH is unsuitable for auxin binding) [42,43,44]. ABP1 involvement in transcriptional developmental responses is still under debate [45,46,47], as it has not been fully demonstrated or refuted, but, since its discovery, knowledge about ABP1 has considerably improved. In fact, it has been reported that the secreted apoplastic portion of ABP1 binds auxin on the cell surface, forming a complex that does not enter the cell but, rather, associates with its specific partner, the transmembrane kinase 1 (TMK1). This signaling tripartite module (ABP1–auxin–TMK1) mediates fast auxin responses, as the regulation of the plasma membrane ion fluxes mediating cell elongation and swelling and/or phosphorylation of a huge number of proteins involved in several cellular processes [48,49,50,51]. As it has been also demonstrated that ABP1 is required for auxin-dependent lobe formation of jigsaw-puzzle leaf pavement cells, this model system was used to investigate a hypothetical involvement of 2,3-MDPU in the spatial coordination of epidermal cells [52,53]. Subsequently, docking simulations were performed between 2,3-MDPU and the crystal structure of ABP1 [54] to unravel possible atomic interactions that could support the biological data.

Second, an interaction with cytokinin oxidase/dehydrogenase (CKX, EC 1.5.99.12), which irreversibly inactivates adenine-type cytokinins by the cleavage of their N^6^-side chains, has been hypothesized. This could be consistent with the inhibition activity of CKX of *Zea mays* (ZmCKX1) [55] and *Arabidopsis thaliana* (AtCKX7) [56] exerted by two urea derivatives, namely 1,3-di(benzo[*d*]oxazol-5-yl)urea (5-BDPU) and 1,3-di(benzo[*d*]oxazol-6-yl)urea (6-BDPU), of which 2,3-MDPU is the lead compound [57]. Therefore, at first, docking simulations were performed with 2,3-MDPU and the crystal structures of ZmCKX1 and AtCKX7 to predict an effective binding mode, then, in vitro enzyme inhibition bioassays were carried out to verify if 2,3-MDPU could inhibit ZmCKX1 as well.

## 2. Results

### 2.1. Adventitious Rooting of Apple Slices

No roots were formed when the slices were cultured in the presence of different NPA concentrations, as expected (Figure 2). When the slices were incubated in the simultaneous presence of 1 μM IBA plus different NPA concentrations, the number of rooted slices significantly decreased as the NPA concentration increased (Figure 2). When the slices were incubated in the simultaneous presence of 1 μM IBA plus 1 μM 2,3-MDPU, the number of rooted slices was significantly higher than that obtained in the presence of 1 μM IBA alone; when different NPA concentrations were added in the presence of the previous compounds, once again the number of rooted slices significantly decreased as the NPA concentration increased (Figure 2).

### 2.2. Adventitious Rooting of Apple Cuttings after Basal NPA Treatment

First type. The basal cut surfaces of apple cuttings were incubated into a micropropagation medium supplemented with NPA for 2 days. When cuttings were then transferred to a hormone-free (HF) medium or into a micropropagation medium supplemented with 2,3-MDPU, adventitious rooting was almost completely absent (NPA-HF-HF and NPA-2,3MDPU-HF conditions in Figure 3A, percentage of adventitious rooting). By contrast, when the cuttings were transferred into a micropropagation medium supplemented with IBA or with IBA plus 2,3-MDPU, adventitious root formation was greatly enhanced, but without any difference between these two rooting-inducing treatments (NPA-IBA-HF and NPA-(IBA+2,3MDPU)-HF conditions in Figure 3A, percentage of adventitious rooting).

Second type. The basal cut surfaces of apple cuttings were incubated into a micropropagation medium simultaneously supplemented with NPA plus IBA or NPA plus 2,3-MDPU, or NPA plus IBA plus 2,3-MDPU [(NPA+IBA)-HF or (NPA+2,3MDPU)-HF or (NPA+IBA+2,3MDPU)-HF conditions in Figure 3B, percentage of adventitious rooting]. Adventitious rooting was achieved only when IBA was present in the medium (NPA plus IBA or NPA plus IBA plus 2,3-MDPU), being slightly higher as to number of roots per rooted cuttings when even 2,3-MDPU was present in the mixture (Figure 3C, number of roots/rooted cuttings).

### 2.3. Adventitious Rooting of Apple Cuttings after Upside Down NPA Treatment

When the cuttings were incubated upside down with the apex immersed in the micropropagation medium supplemented with NPA, their subsequent capacity to form adventitious roots, after transferring them to an HF medium with the correct polarity, was inhibited. A similar result—no adventitious roots—was obtained when the cuttings were transferred to a micropropagation medium supplemented with 1 μM 2,3-MDPU, as this urea derivative is not able to induce adventitious root formation per se. By contrast, when the cuttings were transferred to a micropropagation medium supplemented either with 1 μM IBA or with 1 μM IBA plus 1 μM 2,3-MDPU, adventitious rooting was achieved in both the experimental conditions (90% each, NPA-IBA-HF and NPA-(IBA+2,3MDPU)-HF conditions in Figure 4). Adventitious rooting was enhanced in the presence of the mixture, i.e., IBA plus 2,3-MDPU, as the root number obtained was significantly higher than that obtained using IBA alone (Figure 4).

### 2.4. Histology of Apple Cuttings

Apple cuttings after upside down NPA treatment were used for histological analysis (Figure 5). Cross sections of the bases of 10 µM NPA-treated cuttings, subsequently transferred in the presence of either 1 µM IBA, or 1 µM 2,3-MDPU, or 1 µM IBA plus 1 µM 2,3-MDPU, were analyzed over time. A cross section of an apple cutting consisted of mostly primary tissue; the cambium was fully developed into a continuous ring, with interruptions only at the primary leaf-axillary bud traces, and an initial ring of xylem and phloem was developed. Primary phloem fibers were also visualized (Figure 5A). No significant modifications were observed in untreated cuttings or in cuttings treated with 2,3-MDPU after 8 days. However, an increased rate of cambial, ray parenchyma or phloem initial proliferation was induced in cuttings treated with IBA or IBA plus 2,3-MDPU (Figure 5B,C). Adventitious roots were induced from cambial or phloem initials, and clusters of small clumps of densely stained cells with meristematic traits were observed in the outer phloem of these cuttings (Figure 5B–D). The primordia eventually developed into a functional adventitious root with a vascular connection to the stem base.

### 2.5. Analysis of Arabidopsis Pavement Cell Shape

Exogenous supplementation of 1 μM 2,3-MDPU to the germination medium did not promote an evident morphological modification of the shape of *Arabidopsis* pavement cells. In fact, the number of lobes of cotyledon pavement cells belonging to plants grown in the presence of this urea derivative was similar to that of plants grown under HF culture conditions. On the other hand, when plants were grown in the presence of exogenously supplemented auxin (i.e., 1 μM IBA), the number of lobes was significantly higher than that of plants grown under HF culture conditions. Finally, plants grown in a culture medium supplemented with 1 μM IBA plus 1 μM 2,3-MDPU showed a significantly higher number of lobes, not only than that of plants grown under HF culture conditions, but also than that of plants grown in the presence of 1 μM IBA (Figure 6 and Appendix A, shows the representative phenotypes obtained via the different treatments).

### 2.6. Docking Simulations on ABP1 Receptor

As ABP1 receptor is found in the endoplasmic reticulum and, in smaller quantities, in the apoplastic region [42,43,44,58], it could be reasonably more accessible to 2,3-MDPU binding, therefore it was chosen for our docking simulations.

The ABP1 protein is dimeric in solution as in the crystal [54]. Auxin is quite buried inside the binding pocket, even if its position in the binding cavity is clearly visible, lined by the N- and C-terminal extension and the C and J β-strands of the jellyroll barrel.

To check for the presence of aspecific binding sites for 2,3-MDPU on the protein surface, a blind docking was firstly performed with a coarse mesh grid all around the dimer (see Section 4 Materials and Methods). The larger clusters of docked structures indicated a site (and its symmetric one) located at the edge of the monomers interface, while a second site was found, with a little higher binding energy, in a position close to the entrance of the auxin-binding site.

Subsequently, fine-tuned docking simulations were made around the two sites, and two more defined positions were found. The first, at the monomers’ interface, has an E_i_ = −6.9 kcal/mol and is located between the N-terminal extension of one protein chain and the F and G strands of the other chain. However, the aspecific binding of 2,3-MDPU in this position at the interface is unlikely to interfere with the known auxin pathways of ingress or egress [59].

The second site is more interesting from this point of view. It is at the auxin-binding site entrance, where 2,3-MDPU finds a cleft just in front of the auxin molecule (Figure 7A), with E_i_ = −6.07. In this position, the 2,3-MDPU forms a H-bond between the oxygen of the methylenedioxy ring of MDPU and the Gln52 sidechain nitrogen and several hydrophobic contacts with the protein—in particular, a pi–pi-stacking with Phe149—and with auxin itself (Figure 7B). Noteworthily, Phe149 belongs to the C-terminal region, whose movement was reported to be a gate for auxin binding and which was also hypothesized to be involved in the auxin signaling pathway [59]. On the contrary, in this site 2,3-MDPU does not directly interfere with the three protein tunnels that were identified as egress pathways for auxin in ABP1 (ibid.).

### 2.7. Docking Simulations on CKX Enzymes

The structure of ZmCKX1 (2QKN) in complex with the competitive inhibitor N-phenyl-N′-(2-chloro-4-pyridyl)urea (CPPU), was chosen due to the urea-type chemical structure of the ligand, similar to that of 2,3-MDPU, and previously tested as a reference for docking simulations [57]. In this complex, the preferred orientation shows the 2-chloro-4-pyridyl ring of CPPU in stacking with the isoalloxazine ring of the cofactor flavin-adenine dinucleotide (FAD) [60], while urea backbone nitrogen atoms form hydrogen bonds with the side chain of Asp169, the residue involved in the catalytic mechanism. Another H-bond is formed between Glu381 and the nitrogen of the pyridyl ring of CPPU. The calculated intermolecular interaction energy (E_i_) was −8.49 kcal/mol [57].

Both ZmCKX1 and AtCKX7 (2EXR) crystal structures were used to study, via docking simulations, the possibility of being inhibited by 2,3-MDPU. In both structures, 2,3-MDPU recovers the position of CPPU, behaving in a very similar way to the two previously studied urea derivatives, namely 1,3-di(benzo[*d*]oxazol-5-yl)urea (5-BDPU) and 1,3-di(benzo[*d*]oxazol-6-yl)urea (6-BDPU) [57].

In the ZmCKX1 complex, in the largest docked cluster (44.5% of the conformations), 2,3-MDPU shows the double rings of one methylenedioxyphenyl group in stacking with the FAD isoalloxazine ring (Figure 8), just like CPPU. The H-bond of the urea backbone nitrogen with Asp169 is maintained and another one between Ser456 and the methylenedioxy ring oxygen is formed. The Ei = −7.71 kcal/mol is a little higher than the one of CPPU (−8.49 kcal/mol) and BDPUs (−8.44 and −8.57 kcal/mol) molecules. Interestingly, the second largest cluster (38.5% of the conformations), which is not negligible both for numerosity and intermolecular energy (−7.59 kcal/mol), shows the ligand in a position near the entrance of the binding site, close to the Glu381 that is reported to be a key residue for the enzyme specificity towards some kind of cytokinins [61].

In the AtCKX7 complex, both the best energy and the most numerous docked clusters (together corresponding to 100% of the conformations) show the ligand in almost identical positions, with methylenedioxyphenyl rings in stacking with FAD (Figure 9). The only difference between the two clusters’ conformations is the flipping of the second, not stacked, double-ring group. Also, in the case of AtCKX7, the H-bond between the urea backbone nitrogen of 2,3-MDPU and the Asp162 of the protein (corresponding to Asp169 in ZmCKX1) is conserved, and another H-bond is formed between an oxygen atom of the second methylenedioxy ring and Ser366, that does not correspond to Ser456 of ZmCKX1 but it is positioned at the opposite side of 2,3-MDPU. Interestingly, the position of 2,3-MDPU at the entrance of the binding site is not found for Arabidopsis, probably due to a shift in the sequence that causes the protrusion of the sidechain of Glu367 (analogous to Glu381 of maize) at the opposite side of the α-helix to which they belong.

The E_i_ of 2,3-MDPU in the largest cluster is −8.84 kcal/mol, better than in the ZmCKX1 complex.

### 2.8. In Vitro Inhibition of ZmCKX1

To assess the inhibition of ZmCKX1 by 2,3-MDPU in vitro, we preliminarily evaluated the dependence of the reaction rate on the concentration of its substrate trans-zeatin (tZ), yielding a K_m_ of 6.8 ± 1.3 μM (Appendix A). A tZ concentration of 75 µM was deemed sufficient to reach the V_max_ for subsequent experiments at a fixed substrate concentration. We then measured the apparent Michaelis–Menten kinetics in the presence of 5 and 10 μM 2,3-MDPU. The corresponding Lineweaver–Burk plots exhibited the same intercept on the *y*-axis, a behavior indicative of competitive inhibition (Appendix A). Compound 2,3-MDPU was then assayed as an inhibitor of ZmCKX1 at concentrations ranging from 0 to 333 µM—and at a fixed tZ concentration of 75 µM, yielding an IC_50_ of 19.0 ± 4.3 under our experimental conditions (Figure 10A). A corresponding K_i_ of 1.6 µM was estimated by applying the Cheng–Prusoff equation for competitive inhibition [62]. For comparison, N-phenyl-N′-(1,2,3-thiadiazol-5-yl)urea (thidiazuron, TDZ), a well-known CKX competitive inhibitor [55,63] was assayed under the same conditions, yielding an IC_50_ of 16.9 ± 2.6 μM and a K_i_ of 1.4 μM (Figure 10B).

## 3. Discussion

Adventitious rooting is a complex post-embryonic developmental process affected by exogenous or endogenous factors. Either in native plants affected by catastrophic events, such as flooding or wounding, or in vegetatively propagated plants, adventitious rooting looks like a stress-induced reprogramming of shoot cell fate in which a new root system will necessarily arise from tissues other than the primary root. In this scenario, obtaining a high number of rooted cuttings and/or a high number of adventitious roots per rooted cutting is the goal towards which experimental efforts should be directed.

Here, based on our previous studies, we have analyzed the relationship between auxin, cytokinins and N,N′-bis-(2,3-methylenedioxyphenyl)urea (2,3-MDPU) to better understand the involvement of this synthetic urea derivative in the process of excision-induced adventitious rooting.

### 3.1. Interaction with ABP1

For the first time, an interaction between 2,3-MDPU and the auxin receptor ABP1 has been hypothesized and investigated both from biological and computational perspectives. It is well-known that auxin promotes the lobe formation of jigsaw-puzzle leaf pavement cells and our data are in accordance with this previous information. Furthermore, our results demonstrate that the number of lobes increases significantly when *Arabidopsis* plants are cultured in the presence of IBA plus 2,3-MDPU. Since the association between auxin and ABP1 is essential for the subsequent signal transduction steps leading to lobe formation, we propose that 2,3-MDPU might somehow affect the binding between auxin and its receptor. Docking simulations support this hypothesis, since one of the two aspecific binding sites for 2,3-MDPU on the surface of ABP1 protein that have been found is particularly interesting. This site is located in a cleft at the auxin-binding site entrance where 2,3-MDPU lies in front of an auxin molecule already embedded within its own pocket. The presence of 2,3-MDPU could therefore interfere with auxin–ABP1 binding, possibly enhancing the auxin-driven developmental processes regulated by this receptor.

### 3.2. Interaction with CKX Enzymes

The second kind of interaction investigated was that between 2,3-MDPU and the cytokinin oxidase/dehydrogenase enzyme, again with an experimental and a computational approach. This type of interaction was assumed to result from the chemical structure of 2,3-MDPU, which resembles that of other urea derivatives that inhibit CKX, and in particular from the previously demonstrated inhibition of CKX exerted by the 1,3-di(benzo[*d*]oxazol-5-yl)urea (5-BDPU) and the 1,3-di(benzo[*d*]oxazol-6-yl)urea (6-BDPU), of which 2,3-MDPU is the lead compound [57]. The docking simulations performed with the crystal structures of ZmCKX1 and AtCKX7 predict a binding mode very similar to that of the other urea-type inhibitors, especially CPPU and BDPUs, keeping the same key interactions and comparable energies. The in vitro enzymatic bioassay confirmed that 2,3-MDPU is able to inhibit the ZmCKX1 activity, with an estimated K_i_ comparable to that of TDZ, a potent CKX inhibitor.

Through the biological analyses and the docking simulations performed on both the auxin receptor ABP1 and the cytokinin oxidase enzyme, CKX, we can speculate on the nature of the interaction that can be established between auxin, 2,3-MDPU and cytokinin that could underlie the enhancement of adventitious rooting.

### 3.3. Adventitious Rooting of Apple Slices

This bioassay has been developed to obtain a rapid and reproducible rooting response highly dependent on the medium composition. Concerning hormonal endogenous content, slices are considered as empty explants due to their small thickness (HF-cultured slices do not change their shape and/or morphology during the culture period, rather they turn brown). The experimental system reflects the correct apical–basal orientation of the cuttings, with the medium components flowing down from the apical to the basal side of the slices, as the apical side adheres to the medium surface and the dishes are incubated upside down. In this experimental condition, the supplementation of IBA to the culture medium makes it resemble the apex, and adventitious roots arise from the basal side of the slices (Figure 2). In the simultaneous presence of IBA plus 2,3-MDPU, the number of rooted slices is significantly higher (Figure 2). This enhancement has been already reported as a direct consequence of the simultaneous presence of auxin (IBA) and 2,3-MDPU in the rooting medium [64], but now we demonstrate that auxin availability at a cellular level is essential for adventitious roots to emerge from the slices. Indeed, when increasing concentrations of NPA are added to the culture medium in the simultaneous presence of IBA alone, polar auxin transport is blocked and, consequently, the number of rooted slices is significantly reduced, as expected. Since such a reduction occurs even when increasing concentrations of NPA are added to the culture medium in the simultaneous presence of IBA plus 2,3-MDPU, it can be assumed that 2,3-MDPU neither counteracts nor reverses the blocking of polar auxin transport exerted by NPA. In fact, the rooting-enhancing effect usually exerted by 2,3-MDPU is less noticeable (Figure 2). Therefore, we can also infer that a threshold auxin concentration is required in the simultaneous presence of 2,3-MDPU for an enhancement in adventitious rooting to occur.

Indeed, here we report that 2,3-MDPU binds either to the surface of the auxin receptor ABP1, at the entrance of the auxin-binding site, thus interfering with its release by reducing the flexibility of the C-terminal tail, and, possibly, favoring the association with TMK1; or to CKX, blocking its activity of irreversible inactivation of adenine-type cytokinins. Under these experimental conditions, in which stem slices can be considered as empty explants, it is reasonable to consider that the improvement in the number of rooted slices depends more on the binding of 2,3-MDPU to the auxin receptor ABP1 than on the inhibitory activity of CKX. While the latter is theoretically possible, cytokinins are not present in the culture medium and their endogenous content in stem slices can be reasonably considered close to zero.

### 3.4. Adventitious Rooting of Apple Cuttings

This bioassay was frequently performed, either to verify the rooting capacity of the cuttings, or to analyze the effectiveness of different compounds on the adventitious rooting process. In the latter case, NPA was added to the base of the cuttings before or simultaneously with the adventitious-rooting treatments, as described in Section 4 Materials and Methods. This synthetic inhibitor of polar auxin transport (PAT) not only binds to members of ATP-binding cassette B (ABCB)-type transporters, such as ABCB1 and ABCB19, but even to PIN-formed (PIN) family members, thus blocking auxin efflux movement as recently demonstrated [40,41]. In both cases of basal NPA supplementation, the natural downward auxin flow from the apex is not blocked or even disturbed, but it can be hypothesized that the movement of endogenous auxin efflux from the cells of the basal zone of the cuttings, approximately 1 cm subjected to NPA exposure, is blocked, auxin redistribution fails, and, consequently, adventitious rooting is inhibited. In fact, reversal of this inhibition by NPA is only achieved when cuttings are exogenously supplemented with IBA or IBA plus 2,3-MDPU, as expected. In the latter culture condition, the enhancing effect of 2,3-MDPU in the presence of exogenous IBA is at least partially lost (Figure 3A–C). We can tentatively explain this result by assuming a locally exerted disruptive effect of NPA that (i) blocks the ABCB19 transporter that has been shown to be responsible for the formation of local auxin maxima necessary for adventitious root formation [65], or (ii) might mimic the effect of cytokinin [66]. These effects are in addition to the inhibition of cytokinin dehydrogenase/oxygenase activity by 2,3-MDPU, as indicated by both docking simulations and inhibition of ZmCKX1, so that the local auxin/cytokinin crosstalk is probably highly unbalanced in favor of endogenous cytokinins. Furthermore, the low local auxin concentration and/or redistribution makes the possible interaction between 2,3-MDPU and the apoplastic portion of the auxin receptor ABP1 irrelevant to the responses to auxin that ABP1 naturally mediates [43].

On the contrary, with apical NPA supplementation, PAT is completely blocked; thus, endogenous auxin cannot reach the basal cutting surface and adventitious rooting is inhibited. But, since the basal cells of the cuttings are not affected by apical supplementation of NPA (which probably does not move away from the supplementation zone and accumulates there), they are sensitive to exogenously applied auxin stimulation. Indeed, exogenous auxin supplementation results in adventitious root formation, which is significantly enhanced when IBA plus 2,3-MDPU are applied simultaneously (Figure 4). Under these experimental conditions, the inhibitory activity exerted by 2,3-MDPU on CKX prevents the inactivation of natural adenine-type cytokinins, and consequently, increases the lifespan of cytokinins in plants. Furthermore, the possible interaction between 2,3-MDPU and the apoplastic portion of the auxin receptor ABP1 could modify the auxin-driven responses that ABP1 mediates naturally [43], which could be related to other developmental processes in general, and to adventitious root formation in particular. Thus, cytokinins could locally induce a high amount of cell division of naturally auxin-sensitive cells, leading to the formation of an equally high number of root primordia. The histological analyses confirm that this process really occurs. In this study, a high rate of cambial, ray parenchyma or phloem initial proliferation was induced and clusters of small clumps of densely stained cells with meristematic traits were observed in the outer phloem of the cuttings (Figure 5). These extremely localized clusters of cell divisions resemble those already seen in Gymnosperm hypocotyls when supplemented with IBA plus 2,3-MDPU [32]. Subsequently, root primordia are formed, from which functional adventitious roots develop, according to the typical multiphase progressive process. Interestingly, this situation resembles that recently reported about de novo root formation using *Arabidopsis* whole leaves. In fact, in those model explants, it has been recognized a sequence of four developmental stages: an endogenous micro-callus formation due to intensive cell proliferation, a specification of some of these cells towards the formation of adventitious root founder cells, then root primordia formation and, lastly, adventitious root formation and emergence. These developmental stages are under auxin/cytokinin crosstalk control, as specific auxin- and cytokinin-signaling factors are required for the unfolding of the individual stages [67]. Therefore, it is likely that 2,3-MDPU interferes with the auxin/cytokinin crosstalk, favoring the modulation of an adventitious rooting process, i.e., an auxin-triggered de novo organogenesis [66].

In conclusion, here we shed light on the mechanism related to the peculiar adventitious rooting adjuvant capacity of 2,3-MDPU. While this urea derivative is unable to reverse PAT inhibition caused by NPA, it has a very interesting dual activity. Indeed, it binds to the ABP1 receptor, promoting auxin signal transduction at the cellular level, and it also inhibits CKX activity, increasing the cytokinin lifespan in plants. These two interactions, which may or may not occur simultaneously, may overlap the auxin/cytokinin crosstalk and, probably, explain the adventitious rooting adjuvant activity exerted by 2,3-MDPU under specific experimental conditions. Additional experiments are necessary to elucidate the possible involvement of 2,3-MDPU either in other auxin-driven processes through the ABP1 receptor, like auxin canalization in vasculature formation, or in important cytokinin-driven effects as leaf senescence delay and stress resistance.

## 4. Materials and Methods

### 4.1. Chemicals

The N,N′-bis-(2,3-methylenedioxyphenyl)urea (2,3-MDPU), synthesized as previously reported [30], was of analytical grade. The 2,3-MDPU and the naphthylphthalamic acid (NPA) were dissolved in dimethylsulfoxide (DMSO) and the final concentration of DMSO in the medium did not exceed the one considered toxic (0.2%) [68]. The aqueous solutions of benzylaminopurine (BAP) and indole-3-butyric acid (IBA) were sterilized by filtration using 0.2-μm pore-size sterile disposable filter units (Schleicher and Schuell).

### 4.2. Plant Material and In Vitro Culture Conditions for Apple

In vitro shoot cultures of *Malus pumila* Mill. rootstock M26 were maintained as previously described [30] with minor modifications. The cuttings, deprived of apices, were propagated in tubes on a micropropagation (MP) medium (MS salts, [69], plus 0.4 mg/L thiamine HCl, 0.5 mg/L nicotinic acid, 0.5 mg/L pyridoxine HCl, 100 mg/L myo-inositol, 10 g/L sucrose, 20 g/L sorbitol, 0.65% Phyto Agar (Duchefa), 5.8 pH). The MP medium was supplemented with 1.3 μM benzylaminopurine (BAP) and 0.25 μM indole-3-butyric acid (IBA). After a 6-week incubation stage at 23 ± 1 °C, at a light intensity of 27 μmol m^−2^ s^−1^ under 16 h photoperiod (standard conditions from now on), clusters consisting of 4–6 shoots were formed by axillary branching. The newly formed individual shoots (1.5–2 cm in length, on average) were used either for further micropropagation of for adventitious-rooting experiments.

Rooting experiments included apple slices and cuttings; the variables investigated included the concentrations and combination of NPA, IBA and 2,3-MDPU, as well as the timing and polarity of the application of these substances.

### 4.3. Adventitious Rooting of Apple Slices

Leaves were removed from the M26 shoots and 1 mm thick stem slices were cut using a razor blade [70]. Adjacent slices from the same shoot were distributed over different rooting treatments, to prevent any weak rooting-capacity correlation [71].

-Groups of 25 slices were cultured in Petri dishes with the apical side on a nylon mesh put on an MP medium supplemented with 0.01, 0.1, 1 or 10 μM NPA and with 1 μM IBA as control.-Groups of 25 slices were cultured in the simultaneous presence of 0.01, 0.1, 1 or 10 μM NPA plus 1 μM IBA, while 1 μM IBA alone was used as control.-Groups of 25 slices were cultured in the simultaneous presence of 0.01, 0.1, 1 or 10 μM NPA plus 1 μM IBA plus 1 μM 2,3-MDPU, while 1 μM IBA plus 1 μM 2,3-MDPU was used as control.

All the plates were incubated upside down in the darkness at 23 ± 1 °C for 6 days. Then, the nylon mesh, with the slices attached, was transferred intact to a hormone-free (HF) MP medium and the plates were incubated upside down at standard conditions.

All the experiments were performed in triplicate, repeated twice with similar results and the mean number of rooted slices was calculated after 14 days.

### 4.4. Adventitious Rooting of Apple Cuttings after Basal NPA Treatment

First type. Apple cuttings were incubated with the correct apex–base polarity and their basal portion (approximately 1 cm) was immersed in an MP medium supplemented with 10 μM NPA for 2 days at standard conditions. Then the cuttings were transferred into MP medium in the presence of 1 μM IBA, 1 μM 2,3-MDPU, 1 μM IBA plus 1 μM 2,3-MDPU or in hormone-free (HF) medium as a control condition and incubated at 23 ± 1 °C in the darkness. Again, after 6 days, the cuttings were transferred into MP medium without any further supplementation (HF) and incubated in standard conditions. After 4 weeks from the beginning of the experiment, the number of rooted cuttings were counted. The results are expressed as a percentage of rooted cuttings (i.e., the number of cuttings with emerged adventitious roots on the total microcuttings per treatment). The experiment was repeated three times and 10 cuttings were used in each treatment.

Second type. Apple cuttings were incubated with the correct apex–base polarity and their basal portion (approximately 1 cm) was immersed in an MP medium supplemented with 10 μM NPA plus 1 μM IBA, or 10 μM NPA plus 1 μM 2,3-MDPU or 10 μM NPA plus 1 μM IBA plus 1 μM 2,3-MDPU in the darkness at 23 ± 1 °C. After 6 days, the cuttings were transferred to MP medium without any further supplementation (HF) in standard conditions. After 4 weeks from the beginning of the experiment, the number of rooted cuttings and the number of roots per rooted cutting were counted. The results are expressed as a percentage of rooted cuttings (i.e., number of cuttings with emerged adventitious roots on the total microcuttings per treatment) and as the number of roots/rooted cuttings. The experiment was repeated two times with similar results and 10 cuttings were used in each treatment.

### 4.5. Adventitious Rooting of Apple Cuttings after Upside Down NPA Treatment

Forty apple cuttings were incubated upside down with the apex completely immersed in MP medium supplemented with 10 μM NPA for 2 days at standard conditions. Then the cuttings were transferred with the correct polarity in MP medium in the presence of 1 μM IBA, 1 μM 2,3-MDPU, 1 μM IBA plus 1 μM 2,3-MDPU or in hormone-free (HF) media as a control condition and incubated at 23 ± 1 °C in the darkness. Again, after 6 days, the cuttings were transferred to MP medium without any further supplementation (HF) in standard conditions. After 4 weeks from the beginning of the experiment, the number of rooted cuttings and the number of roots per rooted cutting were counted. The experiment was repeated twice, with similar results.

### 4.6. Histology of Apple Cuttings

For histological analysis of adventitious root formation under upside down NPA treatment, basal 5 mm segments of cuttings were fixed in formalin-acetic acid-alcohol (FAA), dehydrated in a tertiary butyl alcohol series, gradually embedded in paraffin, transversely sectioned at 10 μm thickness with a rotary microtome (AutoCut Microtome 2040, Reichert–Jung, Schönwalde-Glien, Germany) and stained with safranin fast green [72]. Apple cuttings were randomly sampled at the beginning of the rooting experiment (day 0), after 4 and 8 days for each treatment. The histological images were taken in digital form with a Leica digital camera (DMC 2900, Leica Microsystems, Wetzlar, Germany) connected to a Leica DM4000B microscope (Leica Microsystems, Wetzlar, Germany).

### 4.7. Analysis of Arabidopsis Pavement Cell Shape

*Arabidopsis thaliana* ecotype Columbia (Col-0) seeds were surface-sterilized in 70% ethanol for 1 min, followed by 10 min in 50% commercial bleach (equivalent to 2.5% NaOCl), then washed five times in sterile distilled water. They were sown on a germination medium (¼ strength MS salts [68] plus 1% (*w*/*v*) sucrose, 0.8% (*w*/*v*) Phytoagar (Duchefa), 5.8 pH) supplemented with 1 μM IBA or 1 μM 2,3-MDPU or 1 μM IBA plus 1 μM 2,3-MDPU. After a cold treatment at 4 °C for 3 days in the darkness, the plates, containing 25 seeds each, were incubated in a growth chamber at 23 ± 1 °C at a light intensity of 27 μmol m^−2^ s^−1^ under a 16 h photoperiod. Control conditions were performed by seeds grown in hormone-free (HF) germination medium. Seven days after sowing (DAS), the effects of the different supplementations on cotyledon pavement cell shape were evaluated by confocal laser scanning microscopy, as previously described [53], with minor modifications. Arabidopsis cotyledons were mounted in water. Slides were observed with the CLSM system Stellaris 5 (Leica Microsystems, Wetzlar, Germany) using a HC PL APO CS2 63× oil immersion objective (NA 1.4). The acquisition of the autofluorescent signal was carried out, adopting a configuration protocol that required excitation to 405 nm LL (UV laser line) and a spectral detection range of 444–522 nm for cell-shape visualization. The degree of interdigitation in pavement cells was determined by counting the number of lobes of 6 cells, repeating the count for 5 different images collected from cotyledons belonging to different plants (*n* = 30), using ImageJ 1.52h software. The same evaluating procedure was performed for plants grown in the different culture conditions. The experiments were repeated twice, with similar results.

### 4.8. Inhibition Assays of ZmCKX1

ZmCKX1 expression and purification were reported elsewhere [73] and the enzyme assays were carried out as reported before [74,75], with minor modifications. Briefly, the assay mixture contained 1 mM ethylenediaminetetraacetic acid (EDTA), 75 µM dichlorophenolindophenol (DCPIP), 3.3% dimethyl sulfoxide (DMSO) and 4 nM ZmCKX1 in a 100 mM sodium phosphate buffer at pH 7.0. The reaction was started by adding the ZmCKX1 substrate tZ from a 180 mM stock solution in DMSO. The decrease in absorbance at 600 nm associated with the reduction of DCPIP was followed with a Cary 4000 UV-vis spectrophotometer (Agilent technologies, Santa Clara, CA, USA). For the determination of the Michaelis–Menten kinetics, the enzyme was assayed at concentrations of tZ ranging from 5 to 100 µM, either in the absence or presence of 2,3-MDPU at 5 or 10 µM concentration. To evaluate the 2,3-MDPU binding parameters, the inhibitor was pre-incubated at concentrations ranging from 5 µM to 333 µM with ZmCKX1 for 10 min before the addition of 75 µM tZ to the assay mixture. As a positive control, the ZmCKX1 competitive inhibitor thidiazuron (TDZ) was tested under the same experimental conditions in the 5–500 µM concentration range. The TDZ stock solution was at 180 mM concentration in DMSO. The assays were carried out at 30 °C. The IC_50_s were calculated by fitting the experimental points with a hyperbolic equation. The K_i_s were assessed from the IC_50_s by applying the Cheng–Prusoff equation for competitive inhibition [62], considering an experimentally determined K_m_ for tZ of 6.8 µM, close to the value of 14 µM reported in the literature [76]. All plots and data analysis were performed using the section OriginPro, Version 2023 (OriginLab Corporation, Northampton, MA, USA).

### 4.9. Molecular Docking Simulations

The docking simulations were performed with the Autodock 4.2 software package, after preparing the input files with the aid of the AutoDockTools 1.5.7 (ADT) interface [77]. The Lamarckian Genetic Algorithm [78] was used for all docking calculations.

A conformational cluster analysis was performed on the docked conformations [78] with a rmsd cluster threshold of 2 Å, and structural analysis of the binding modes was made using ADT 1.5.7, VMD 1.5.3 [79] and Swiss-PdbViewer 4.1.0 [80] softwares.

#### 4.9.1. ABP1 Receptor

The only crystal structure of an ABP1 receptor available in the Protein Data Bank [81] was the one from *Zea mays*, sharing a 63% sequence identity with *Arabidopsis thaliana* (the model plant used in the pavement cell-shape bioassay), and reaching 88% if we also consider conservative residues. Even if the structure of *Arabidopsis thaliana* is available as an AlphaFold prediction, for a docking simulation—in particular a blind docking—the use of a crystal structure is recommendable. Therefore, the ABP1 dimeric structure, complexed with auxin and zinc and containing a glycosylated amino acid (Asn95), was chosen for docking simulations (PDB ID: 1LRH [54]). The A–D chains were extracted from the crystal, keeping the auxin molecules, the zinc atoms and the glycosylated residues, and deleting only water molecules. By means of ADT 1.5.7 software [77], the atom types of non-protein molecules were checked and Gasteiger partial charges were assigned [82]. Histidine residues were kept in the neutral form.

To find aspecific binding sites on the protein surface, a blind docking with a coarse mesh grid 0.653 Å spaced was performed. The grid box was 104 × 82 × 94 points large, comprehending the whole dimer surface. 1000 runs were performed, each with 25 × 10^6^ energy evaluations, 250 individuals in the initial population, and 27,000 generations. The cluster analysis allowed the study to identify the two most favored binding regions on the surface: one at the dimer interface, the other one close to the auxin-binding site entrance (see the Section 2 for details). Both sites were selected for a deeper analysis, performed around the two identified regions new docking simulations with a finer mashed grid (0.375 Å) and smaller boxes (74 × 56 × 74 points for the site at the interface and 54 × 64 × 66 points for the site close to the auxin entrance path). Again, 1000 runs were performed for each box, with 5 × 10^6^ energy evaluations, 200 individuals in the initial population, and 27,000 generations.

#### 4.9.2. CKX Enzymes

For docking simulations with cytokinin oxidase/dehydrogenase enzymes, both *Zea mays* (ZmCKX1) and *Arabidopsis thaliana* (AtCKX7) crystal structures (PDB ID: 2QKN [55] and 2EXR [56], respectively) were selected and prepared as reported in our previous work [57]. The same grid parameters were also set following that previous research.

#### 4.9.3. MDPU Ligand

The structure of 2,3-MDPU for docking simulations was built by means of a PRODRG server [83] and minimized by Avogadro 1.2.0 Software [84] with the MMFF94 force field [85]. Gasteiger partial atomic charges were used for docking [82]. Only the two torsions around the bonds between the N atoms and the aromatic substituents were kept free to rotate.

### 4.10. Statistical Analyses

#### 4.10.1. Adventitious Rooting of Apple Slices

Analyses of variance (ANOVA) was performed and the significantly different values among the mean number of rooted slices were identified via a post hoc Duncan’s test (*p* ≤ 0.01), using the statistical software package IBM SPSS 26 (Figure 2).

#### 4.10.2. Adventitious Rooting of Apple Cuttings after Upside Down NPA Treatment

The mean number of roots per rooted cuttings was calculated and the significant difference was determined using Student’s *t*-test (*p* ≤ 0.01) (Figure 4).

#### 4.10.3. Analysis of *Arabidopsis* Pavement Cell Shape

Analyses of variance (ANOVA) was performed and the significantly different values among the mean number of lobes were identified by a post hoc Duncan’s test (*p* ≤ 0.05), using the statistical software package IBM SPSS 26 (Figure 6).

## Figures and Tables

**Figure 1 plants-12-03610-f001:**
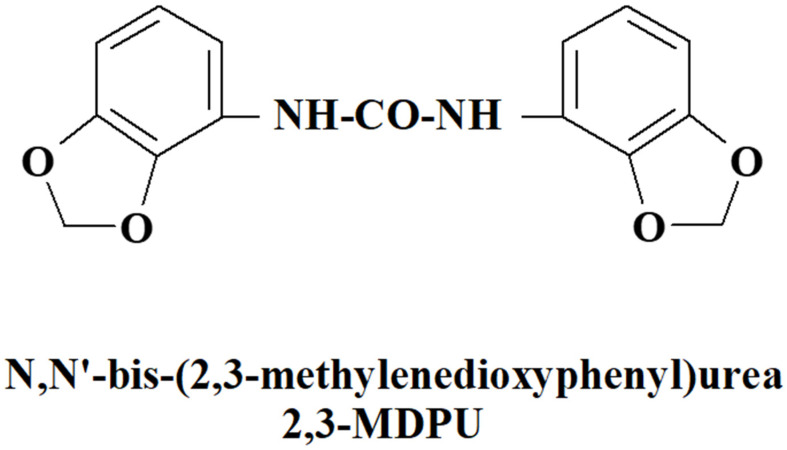
Molecular structure of the urea derivative used in this study.

**Figure 2 plants-12-03610-f002:**
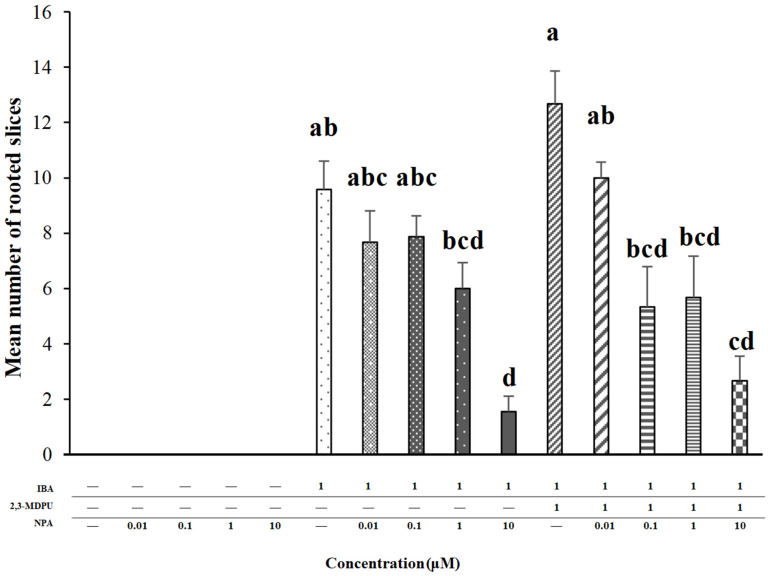
Effect of increasing concentrations of NPA alone or in the simultaneous presence of 1 μM IBA, or in the simultaneous presence of 1 μM IBA plus 1 μM 2,3-MDPU, on adventitious rooting of apple stem slices. The mean number of rooted slices was calculated after 14 days of culture. Means ± SE, different letters indicate significant differences among treatments according to Duncan’s test (*p* ≤ 0.01) (*n* = 150).

**Figure 3 plants-12-03610-f003:**
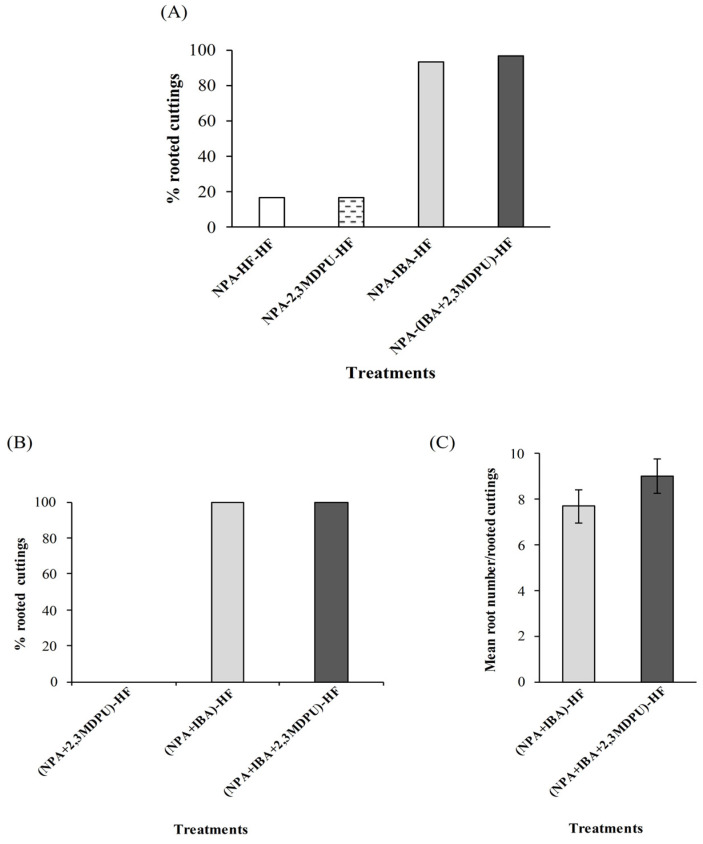
(**A**) Effect of the first type of NPA basal treatment on adventitious rooting of apple cuttings, results are expressed as percentage of rooted cuttings; (**B**,**C**) effect of the second type of NPA basal treatment simultaneously supplemented with IBA or 2,3-MDPU, or IBA plus 2,3-MDPU, results are expressed as percentage of rooted cuttings in (**B**), as mean root number per rooted cuttings in (**C**) (*n* = 30).

**Figure 4 plants-12-03610-f004:**
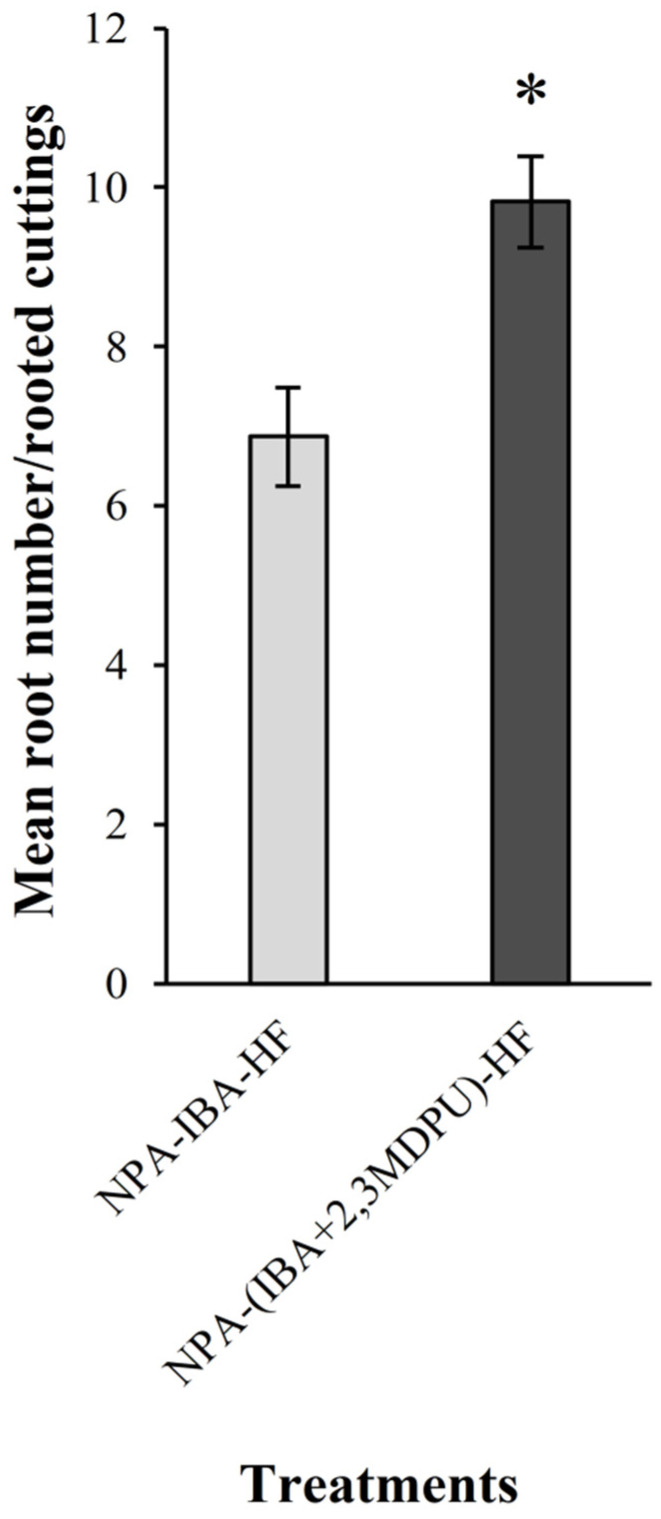
Effect of NPA apical treatment on adventitious rooting of apple cuttings. The mean number of roots was calculated after 4 weeks of culture. Means ± SE, column labeled with asterisk is significantly different according to Student’s *t*-test (*p* ≤ 0.01) (*n* = 20).

**Figure 5 plants-12-03610-f005:**
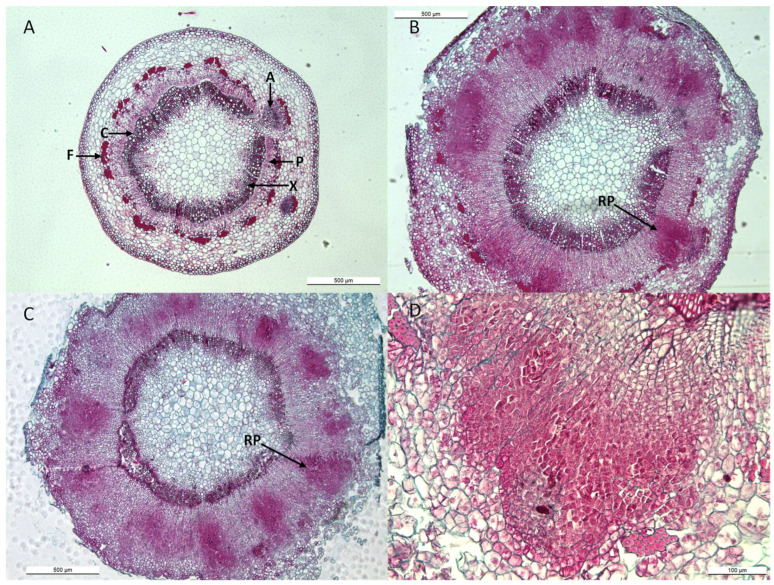
Histological analysis during the adventitious root formation in apple cuttings after upside down NPA treatment. (**A**) Cross section of an apple cutting at the excision (time 0). (**B**) Cross section of the base of a 10 µM NPA (apical supplementation) plus 1 µM IBA-treated cutting after 8 days. (**C**) Cross section of the base of a 10 µM NPA (apical supplementation) plus 1 µM IBA plus 1 µM 2,3-MDPU-treated cutting after 8 days. (**D**) Detail of a root primordia induced in the base of a 10 µM NPA (apical supplementation) plus 1 µM IBA plus 1 µM 2,3-MDPU-treated cutting after 8 days. (A) axillary bud, (C) cambium, (F) phloem fibers, (P) phloem, (RP) root primordia, (X) xylem.

**Figure 6 plants-12-03610-f006:**
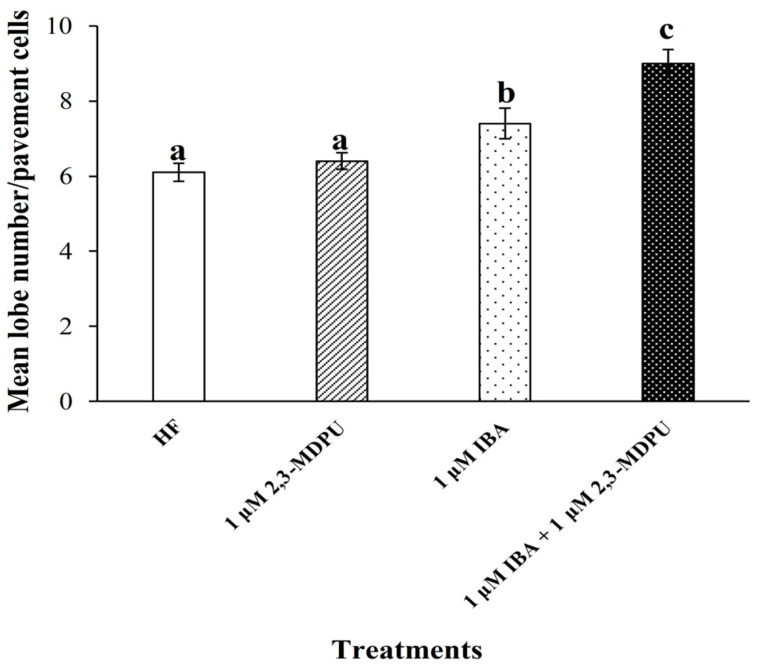
Effect of 1 µM IBA, 1 µM 2,3-MDPU or 1 µM IBA plus 1 µM 2,3-MDPU on mean lobe number in Arabidopsis cotyledon pavement cells. Means ± SE, different letters indicate significant differences among treatments according to Duncan’s test (*p* ≤ 0.05) (*n* = 60).

**Figure 7 plants-12-03610-f007:**
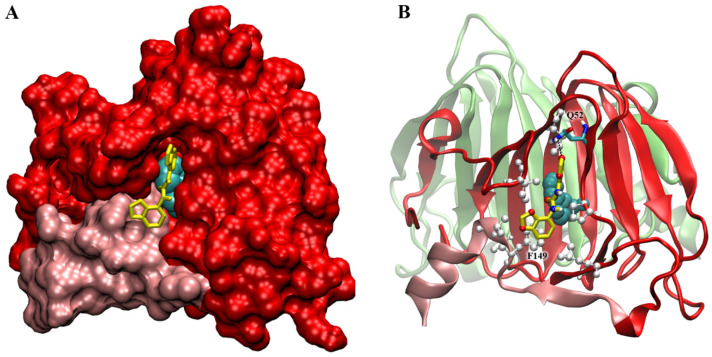
2,3-MDPU docked at the surface of ABP1 receptor. The “second” site, at the auxin-binding site entrance, is shown. (**A**) The ligand is inserted in a cleft in the protein surface (in red) in front of the auxin molecule (in cyan and VdW representation). The surface of the C-terminal region of the monomer is highlighted in pink. Only one monomer is shown for clarity. (**B**) The same ligand position in the protein in a cartoon representation (here also the second monomer is shown, in lime), to highlight the hydrophobic residues around 2,3-MDPU (see Phe149 in stacking) and Gln52 forming a H-bond (dashed line) with the ligand.

**Figure 8 plants-12-03610-f008:**
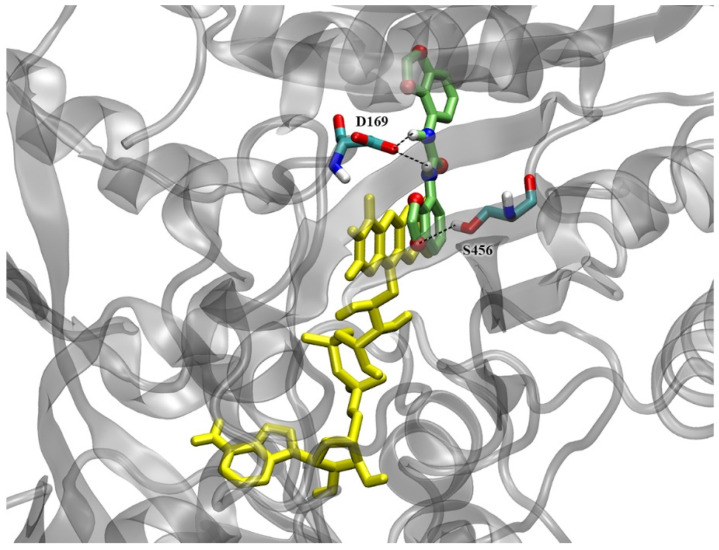
2,3-MDPU (in lime, with polar atoms colored by type) docked inside the ZmCKX1 binding site. The representative (best energy) conformation of the largest cluster is shown. The stacking with FAD rings (in yellow) is evident. Residues forming H-bonds (dashed lines) with 2,3-MDPU are highlighted and colored by atom type.

**Figure 9 plants-12-03610-f009:**
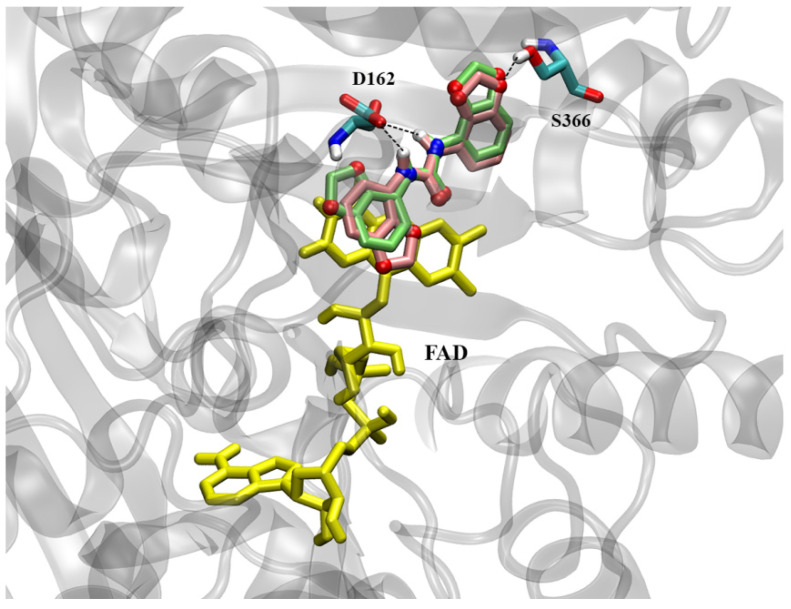
2,3-MDPU docked inside the AtCKX7 binding site. The representative conformations of the two clusters are shown in lime and pink, with polar atoms colored by type. The stacking with FAD rings (in yellow) is evident. Residues forming H-bonds (dashed lines) with 2,3-MDPU are highlighted and colored by atom type.

**Figure 10 plants-12-03610-f010:**
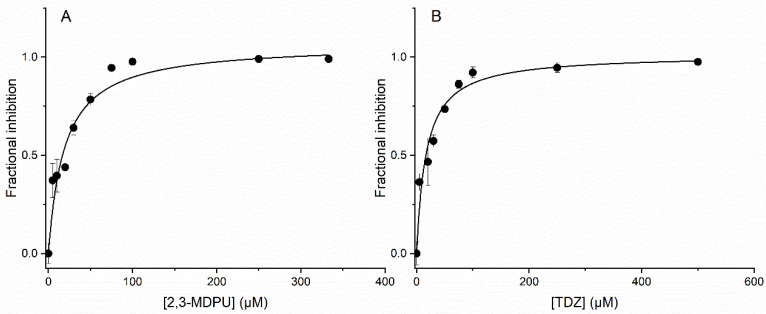
Inhibition curves of ZmCKX1 at concentrations of (**A**) 2,3-MDPU, ranging from 0 to 333 μM and (**B**) TDZ, ranging from 0 to 500 μM. The experimental points (black circles) are the mean of three independent measurements. Error bars indicate ±SD. The solid lines are the fitting to binding isotherms. The r^2^ are 0.90 and 0.96, for 2,3-MDPU and TDZ, respectively. All measurements were carried out at 30 °C.

## Data Availability

Not applicable.

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
