# Peer review of "New Insights into the Enhancement of Adventitious Root Formation Using N,N′-Bis(2,3-methylenedioxyphenyl)urea"

_plants, 2023, doi:10.3390/plants12203610_

Round 1

Reviewer 1 Report (New Reviewer)

The manuscript presents in detail how N,N' bis --(2,3 methylenedioxyphenyl)urea can help to root apple stem segments. The manuscript was well written, but some details must be considered before final acceptance. In many parts of the text, the unit of concentration (micromolar) is incorrectly typed. Figure 4 presents problems regarding axis values. Unity 2.6 and 2.7 looks more like the discussion. I understand that the authors tried to give background on the subject, but this should be done in the introduction. I recommend deleting the first two paragraphs of the discussion, as they don't add much.

Author Response

Response to reviewer 1

We are very grateful to reviewer 1 for the consideration of our research and for the comments submitted to improve the manuscript.

  • The unit concentration (micromolar) is incorrectly typed due to a mistake coming from the editorial process. I will personally edit in the correct font all the wrong printed units.
  • The title of y-axis of the graph in Figure 4 has been changed and a much more comprehensive title has been added in x-axis, hoping to improve the understanding of the results shown.
  • About Unit 2.6 and 2.7: the information about the background on the subjects was included at the beginning of the units and not in the introduction as not to make it too heavy, but, in order to follow the suggestion, same of them has been deleted while some others shortened.
  • The first part of the discussion has been shortened, as suggested.

We hope that the new version of the manuscript could be accepted for publication.

Reviewer 2 Report (New Reviewer)

The authors  proposed a molecular mechanism by which  MDPU in stimulates the formation of adventitious roots. The hypothesis of binding with the ABP1-auxin receptor and to CKX was tested  by several methodological approaches, that are suitable for the intended purposes. However, I detected some experimental weakness in the MS, especially in the number of repetitions,  in the presentation of results and in the statistical analysis, as listed below.

Number of repetitions: The authors repeated most of the experimental series only twice.  With that small number, it would not be possible to carry out the appropriate statistical analysis or the t-Student test, much less the ANOVA. For the t test, the minimum number of repetitions should be three,  with plant material processed from the beginning, preferably on different days. In addition, to use ANOVA with Ducan post hoc testing, the data must first pass the variance homogeneity test, which cannot be done with such a small number of repetitions. In the captions of the Figures,  the authors must  mention the exact number of repetitions (n) performed in each test.

Results presentation:

Results of Figure 2: In this experimental series, there are three independent variables and the correct  statistical analysis is not to compare just one series independently of the other (with or without MDPU) using one-way ANOVA, but ANOVA with multiple variables. For this, however, it is necessary to have a  higher number of repetitions.

Results in Figure 3: Panels (a) and (ba) are shown as a percentage of the highest value obtained. This is also not adequate, as the percentage is always in relation to the control value, in this case it would be the NPA-HF-HF, since the treatment with IBA and the MDPU are the treatments. In anyway, the correct mode of presentation is the experimental values obtained (number of adventitious roots), instead of relative values.

   Results of Figure 4: the Figure in my version came without information on the x axis, and also without annotation if there was statistical significance. The stimulus caused by the presence of the MDPU is small and without a reasonable number of repetitions that pass through the homogeneity test, it cannot be categorically stated that the stimulus is real.

Results Figure 6: It is known that molecular docking simulation assays are a powerful tool to detect interaction of substances with proteins, or to find enzymatic inhibitors for the most diverse applications. However, a good interaction does not always translate into expected effects in in vivo assays. Thus it is always necessary to perform a biological test, as was done in the case of the CKX enzyme. In the case of the interaction with the ABP1 protein, the  authors presented as experimental evidence of the interaction of MDPU with this protein the Analysis of Arabidopsis pavement cell shape, which has already been demonstrated to be a phenomenon created by the interaction of auxin with the ABP1 protein. This  series is essential to correct conclusion, however, besides clarifying the number of repetitions made  and carrying out adequate statistical analyses, the authots shoud include representative photos of the increase in the number of lobes indicated by arrows. These photos may be included in supplementary material.

Figure 10. The MPDU experimental curve analysis is not adequate. The theoretical curve using the assumption that MPDU is a competitive inhibitor does not fit well with the experimental curve. What was the values of   standard error of the estimate (s) and the coefficient of determination (r2)? Many substances, despite being structurally similar to the substrate, or to other known inhibitor do not necessarily exert the competitive inhibition or the same kind of inhibition of structurally similar compounds. This enzyme kinetics analysis needs to be performed adequately.

Discussion:

Without the replies to criticisms raised above with the pertinent modifications in the MS, I  cannot be able to evaluate the Discussion section. The results and statistical analysis should be robust and reliable, so that it can be categorically stated that MPDU stimulates the formation of adventitious roots by acting in association with auxin through interaction with the ABP1 protein and/or inhibiting CKX.

Author Response

Response to reviewer 2

We are grateful to the reviewer 2 for acknowledging that “the many methodological approaches used in our research are suitable for the intended purposes”. In the meantime, we regret that all the information provided was not sufficient for understanding the results obtained.

About number of repetitions:

  • The experiments were sometimes done only twice, as the results were absolutely similar. To emphasize it, this information has been added at the end of Units 4.3, 4.4, 4.5. Obviously, the experiments have been performed in different days, starting from different plant material each time (it could not be otherwise).
  • The exact number of repetitions (n) performed in each test has been added in the captions of the figures, as requested.
  • Starting from our experience dealing with adventitious rooting experiments, regarding Student’s t test and the results shown in the graph of Figure 4, we think that the total number of repetitions, 20 for each treatment, is sufficient to carry out an appropriate statistical analysis.
  • Regarding ANOVA analyses, we assure you that all the preliminary tests that allowed us to perform the ANOVA have been carried out (and successfully passed), as, contrary to what you think, in our opinion the number of repetitions is not small.

About presentation of results:

  • The graph of Figure 2 shows the results obtained when apples slices were supplemented with NPA, IBA, 2,3-MDPU alone or in mixtures, in terms of mean number of rooted slices. The slices treated with NPA did not root, as expected, thus these results were discarded and only the others were analysed with ANOVA. There are not 3 independent variables, but 10 different conditions, i.e. different treatments, in which adventitious roots were formed. The effects of these 10 different treatments were compared with ANOVA. Each plate contained 25 slices, three plates for each treatment per experiment, two independent experiments were done. Do you really think that number of repetitions is small?
  • I am very sorry but the editor provided you with an old version of the manuscript. In the new one, that you should have received to correctly review the manuscript, Figure 3 has panel A), B) and C). By the way, the data shown in panels A) and B) are not the “percentage of the highest value obtained”. They are not “relative values”, as they are not in relation to the control value, in fact, control value is shown in the graphs. Very simply, we decided to express the number of cuttings with emerged adventitious roots on the total microcutting per treatment as a percentage. As an example: 20 cuttings are supplemented with a rooting treatment and adventitious roots emerge from 3 of them. How it is possible to express this result? As 15% of rooting. This is a usual and very common way to quantify the rooting capacity, there are no artifacts, there is no strange transformation formula, it’s a proportion.

Hoping to improve the understanding of the results shown, the title of y-axis of the graphs in panel A) and B) of Figure 3 has been changed and a much more comprehensive title has been added in x-axis; the caption of panels A) and B) of Figure 3 has been changed. In Materials and Methods, in Unit 4.4, a sentence has been added, in order to better explain the way the results are expressed.   

  • Again, the editor provided you with a version of the manuscript without the annotation of significance. It’s not our fault, it’s a editing mistake. The title of y-axis of the graph in Figure 4 has been changed now and a much more comprehensive title has been added in x-axis, hoping to improve the understanding of the results shown. Starting from our experience dealing with adventitious rooting experiments, regarding Student’s t test and the results shown in the graph of Figure 4, we think that the total number of repetitions, 20 for each treatment, is sufficient to carry out an appropriate statistical analysis. The first publication regarding 2,3-MDPU and its involvement in adventitious rooting is from 2001 and the stimulus caused by 2,3-MDPU is real since then (please, have a look at the literature).
  • The exact number of repetitions (n) has been added in the caption of Figure 6 and representative photos of the increase in the number of lobes has been included as supplementary material (Figure S1), as requested.
  • As for the observations regarding Figure 10, the reviewer is right: we assumed a competitive inhibition based on published results on analogs. To confirm that this is indeed the inhibition mechanism: i) we measured the dependence of velocity on substrate concentration (now figure S2); ii) we measured the apparent Michaelis-Menten parameters in the presence of two inhibitor concentrations (Figure S2); iii) we added experimental points to Figure 10 to improve the quality of the fitting. We added the relevant pieces of information required by the reviewer about the data analysis. Because of the additional data and new analysis, the resulting Ki are slightly different than those reported in the original manuscript.

We hope that the new version of the manuscript could be accepted for publication.

This manuscript is a resubmission of an earlier submission. The following is a list of the peer review reports and author responses from that submission.

Round 1

Reviewer 1 Report

Ricci et al. aim to describe in the present manuscript how the urea-derived compound MDPU enhances the regulation of the root adventitious root formation. However, in the present paper is exclusively a description of the effect of the MDPU compound in the formation of adventitious roots in auxin-transport impairment conditions (NPA). They conclude that MDPU enhances the formation of these roots only when the auxin transport is not severely affected. Additionally, the show a histological study and a docking experiment. Using as only result the docking experiment (and mentioning other publications) they built the conclusion that MDPU could have a dual role binding CKX and ABP1 proteins. Although the work could be interesting it is based in non-acceptable overstatements. To demonstrate that MDPU enhances the formation of adventitious roots, further experiments with CKX or ABP1 mutants are necessary, or to perform enzyme/receptor activity assays.

Major concerns:

1) In the title it says that the MDPU “enhances adventitious root formation by modulating auxin and cytokinin stimuli”. This is a clear overstatement since not a single experiment in the manuscript that proves that CK activity is modulated by MDPU.

2) On several occasions the authors claim that cytokinin induces cell division. However, it is widely proved that in roots, cytokinin induces cell differentiation and impairment of the cell division (i.e., Dello Ioio 2007, Current Biology). How the authors explain their claim of MDPU impairing CKX function and therefore increase of cytokinin activity?

3) Table 1. It is important to show statistics comparing NPA+IBA+MDPU versus NPA+IBA to see whether MDPU is really doing some effect or can be considered a normal deviation of the median.

4) Figure 2 and 3. Are missing statistics, how to compare them? There are no error bars. Are they more than 1 biological replicate?

5) Figure 4. It would be important to mark in the pictures the different areas of the root to see the differences they claim in the text (Line 160-173).

6) The docking experiment for ABP1 is made using the Zea mays structure, but the other experiments are done in Malus pumila. Degree of similarity among them?

Minor concerns:

1) Figure 1. The quality is very poor and would be beneficial to include the structure of auxin and cytokinin to show how similar are they to MDPU.

2) MP (line 128 and more) and HF (line 129 and more) are mentioned in the text, but the meaning of the acronym is only explained at the end of the manuscript (in methods).

3) Line 150: “data not shown”. It is not acceptable in science nowadays to use this expression, so is better either to show the data or to delete the sentence.

3) Lines 62-63: This statement needs a reference.

4) Line 25: Space after dot.

Reviewer 2 Report

The manuscript 'N,N’-bis-(2,3-methylenedioxyphenyl)urea Enhances Adventi-2 tious Root Formation by Modulating Auxin and Cytokinin 3 Stimuli' provides short informations about potential action of 2,3-MDPU in micropropagated tissue. Before accepting this paper I recommend few modifications in the text. Some sentences or phrases are highlighted in the text, althought I suggest to check all the text once again. 

Sections 'Introduction' and 'Conclusion' must be improved. Also the presentation of the results (statistics) needs some changes.

Furthermore, I suggest to prepare the scheme of used procedure (exposure to PGR) to improve understanding for the readers. Now it is confusing.

After referring to my comments, I think the paper is suitable for publication.

Reviewer 3 Report

Dear authors, 

the work is interesting with lots of work done. However, the paper is written with not enough care. Misleading citations, figure designations etc.

I suggest careful major revision before re-submission.

Table 1: there needs to be statistisc and SD or SE.

Lines 119 - 122. Conclusions are not true/significant without stats. Data make no clear conclusion.

Table 2 and lines 127 - 140: I do not understand to designation of figures in fig. 2. I see Fig. 2a, 2b, 2c, nothing else like 2A, 2B or 2Ba, 2Bb (line 140). In text there is not 2c.

Again, Fig.2 is missing SD/SE and stats.

Fig.3. (line 157 - 158) ,, Column labelled with asterisk is significantly different,, There is no astrisk in Fig. 3.

Fig. 4. Designation of,, a) axillary bud, c) cambium, f) phloem fibers, p) phloem, rp) root primordia, x) xylem,, I am not sure if I can see all these in figure 4. The designation in Fig. 4. should be bigger, more visible, and maybe with dark arrow showing the exact place of designation.

Line 182 and further: citation 44 in the text is supposed to be 43 (probably).

Anyway, I think docking studies cannot inform about the inhibition ability of the compound. I strongly suggest to test the 2,3-MDPU molecule for ZmCKX1 inhibition and possibly with other enzymes which are accessible (at least AtCKX2). Then the story will be more significant, as a docking studies are really weak to make some conclusions.

Figures 5 and 6 shall be better presented, so the interactions are visible.

Line 235: citation 53 is not related to the text.